# Does School-Level Instructional Quality Matter for School Mathematics Performance? Comparing Teacher Data across Seven Countries

Xin Liu [1,*], Martin Valcke [1], Kajsa Yang Hansen [2] and Jan De Neve [3]

1   Department of Educational Studies, Faculty of Psychology and Educational Sciences, Ghent University, Henri Dunantlaan 2, 9000 Ghent, Belgium; martin.valcke@ugent.be

2   Department of Education and Special Education, Faculty of Education, University of Gothenburg, Västra Hamngatan 25, 40530 Gothenburg, Sweden; kajsa.yang-hansen@ped.gu.se

3   Department of Data Analysis, Faculty of Psychology and Educational Sciences, Ghent University, Henri Dunantlaan 1, 9000 Ghent, Belgium; jan.deneve@ugent.be

*   Correspondence: xin.liu@ugent.be

**Abstract:** Based on the TALIS 2013 and PISA 2012 linkage data, we examine the measurement properties of school instructional quality and study its relationship with mathematics performance, considering school context characteristics (school composition, teacher–student relationship, and teacher qualifications). The study adopts a cross-country perspective. In five of the seven countries, a three-dimensional framework has been confirmed to study mathematics instructional quality (disciplinary climate, supportive climate, and cognitive activation). As a common factor, disciplinary climate explains the variation in school mathematics achievement in four countries. The key is the interaction with socioeconomic status. Schools composed of students with favourable socioeconomic backgrounds reflect a disciplinary climate conducive to learning. Schools consisting of students with low socioeconomic backgrounds benefit more from a supportive climate, contributing to the reduction in the achievement gap. Schools with harmonious teacher–student relationships reflect differential effects on mathematics performance of schools consisting of students from lower- and higher-socioeconomic status families. Low-SES schools are more likely have less academically qualified teachers. School collective teacher qualification seems not directly related to school mathematics performance, but disciplinary climate mediates this link. Consistently, schools composed of students from high-socioeconomic status families tend to perform better.

**Keywords:** mathematics instructional quality; school mathematics performance; school context characteristics; measurement invariance; cross-country comparative study

## 1. Introduction

School effectiveness research attempts to identify the role of school characteristics in explaining variance in educational outcomes [1–5]. Examples of such characteristics are teacher–student relationships, instructional quality, or school composition. Given the fact that instructional quality reflects teacher behaviour in the classroom and positively impacts students' learning outcomes [6–9], studies show a significant amount of variation in instructional quality between schools [10–13]. However, there are insufficient studies highlighting the relationship between instructional quality and academic performance in terms of school context features such as school composition or collective teacher qualifications or teacher–student relationship [14–16]. For example, some studies indicate that schools serving a relatively large percentage of low-SES students tend to employ less qualified teachers (e.g., teacher educational background, work experience) than schools serving more advantaged students [17–19], thereby affecting teaching behaviours and learning outcomes [20–22]. Some studies underline the positive mechanism between school composition and teacher–student relationships [23–25]. This implies that positive interpersonal

relationships facilitate effective learning, which directly and indirectly enhances academic performance [24,26–28].

Drawing on the unique features of two international large-scale assessments, the Teaching and Learning International Survey (TALIS) and the Programme for International Student Assessment (PISA), this paper applies the linkage dataset of TALIS 2013 and PISA 2012. Linking data from TALIS 2013 and PISA 2012 helps to examine the impact of mathematics teachers' instructional quality on learning performance in the school context [29]. While several authors have already relied on the TALIS and PISA linkage data in studying mathematics performance [30,31], there has been a lack of focus on mathematics teachers. The focus on particular subjects is important since earlier studies point at differences in conceptions of instructional quality across subjects [32,33]. Moreover, studies point at differences in perceptions about instructional quality when looking at teachers and students [34–36].

Therefore, in the current study, we want to establish a more nuanced view of the relationship between mathematics teachers' instructional quality and school mathematics performance. We first explore the factor structure of school mathematics instructional quality building on TALIS 2013 mathematics teachers' data from seven countries. When implementing TALIS 2013, participating countries could opt for an extra PISA-related survey that required all mathematics teachers to participate in the Mathematics Teacher Questionnaire. This extra option is labelled as the TALIS-PISA Link. More specifically, mathematics teachers were considered to be a subsample of teachers who did teach students in schools that participated in PISA 2012. Eight countries opted to participate in this extra survey: Australia, Finland, Latvia, Mexico, Portugal, Romania, Singapore, and Spain. The current study is a follow-up to other studies in which we applied linkage data. In the previous study, the measurement model for Mexico was different from any other country. In view of consistency, Mexico was therefore not included in the analysis in the present study. Thus, seven countries were selected in the following analysis—Australia, Finland, Latvia, Portugal, Romania, Spain, and Singapore. Secondly, we tested the measurement invariance of school mathematics instructional quality when comparing the countries. Thirdly, while considering socioeconomic status, a two-level model was adopted to study associations between performance and instructional quality. At the student level, individual socioeconomic status from PISA 2012 was used to explain the variance in student performance. At the school level, we focused on evaluating the relationships between mathematics instructional quality from TALIS 2013 and school mathematics performance derived from PISA 2012, taking into account the school's profile of socioeconomic status, teacher qualification, and the teacher–student relationship. The reason for investigating instructional quality building on mathematics teacher data at the school level is that TALIS and PISA share one key variable: school ID. This helps to combine data from both mathematics teachers and students from the same schools. Our analyses provide insight into the variation in school effectiveness in mathematics performance that can be explained by mathematics instructional quality and how socioeconomic gaps might be reduced. A cross-country comparative perspective provides insights into diverse cultures and reflects the mathematics teaching activities of the school system.

## 2. Literature Review

### 2.1. The Construct of Instructional Quality

Instructional quality reflects teacher behaviour in the classroom and is consistently linked to cognitive and non-cognitive student outcomes [6,37–39]. Multiple studies underline the multi-dimensional nature of instructional quality [9,40–42]. A first basic framework puts forward three instructional quality dimensions: classroom management, supportive climate, and cognitive activation [39,43]. Classroom management refers to the effective use of learning time and teachers' behaviour in dealing with disciplinary disturbances in the class [39,44]. Supportive climate describes the extra help, useful feedback, and emotional support provided by teachers in the student learning process [39,43]. Cognitive activation

emphasises invoking knowledge integration and cognitive engagement in the problem solving and decision making of challenging tasks [39,45].

In educational effectiveness models, other instructional quality dimensions are being stressed that go beyond the three-dimensional framework [39]. For example, the Dynamic Model of Educational Effectiveness puts forward eight dimensions: orientation, management of time, questioning, classroom climate, structuring, application, teaching–modelling, and assessment [45,46]. The 7Cs Framework of Teaching Effectiveness consists of seven dimensions: classroom management, care, confer, captivate, clarify, consolidate, and challenge [47,48]. In international large-scale studies—such as TIMSS, PISA, and TALIS—the dimensions of cognitive activation, support climate, classroom management, and clarity of instruction are consistently used to map instructional quality [9,49,50].

The differences in conceptualising the construct of instructional quality seem to reflect differences in (1) whose perspective is being adopted, (2) whether a general or subject perspective is being employed, and (3) whether cultural differences between countries have been considered. Firstly, instructional quality dimensions seem to differ when building on data from teachers or students [34,35,51]. For instance, the dimension of student-oriented instruction is captured when building on student ratings [49,52]. Furthermore, dimensions of clarity of instruction, feedback, and assessment are put forward as critical dimensions when building on teacher and school evaluator data [50,53–55]. Secondly, specific school subjects seem to put forward other teaching demands, resulting in other instructional dimensions being emphasised [32,33,56]. For example, in mathematics education, studies identified the dimensions of student-oriented instruction from student data and classroom disciplinary climate from both teacher and student perspectives [49,52]. Thirdly, researchers stress that participants from different national and cultural backgrounds might interpret the concept differently [57], resulting again in different frameworks of instructional quality being characterised [40,52,58]. Most international large-scale studies adopt a generic conception of instructional quality that is considered to be independent of country-specific school characteristics. It is possible that this assumption may neglect country-specific interpretations of the latent structure. This calls for cross-cultural validation studies of the measurement properties and measurement invariance tests of the construct of school teaching quality. However, such studies are rare [9,40,41,59].

The former brings us to the specific focus adopted in the present study. We build both on teacher and student ratings of instructional quality. Next, we look at instructional quality from the mathematics subject point of view, and lastly, we check the measurement properties of instructional quality across different countries.

### 2.2. School-Level Instructional Quality and Academic Performance

Studies measuring instructional quality are typically conducted at the teacher/classroom level. Few studies address the nature of instructional quality from a school-level perspective [53,60]. The findings from Wenger et al. [11] and Hill et al. [10] demonstrate that schools differ systematically in their instructional quality. Moving to the school level has specific consequences. Marsh et al. [61] stated that a construct might have a distinct meaning when aggregated at higher-level units. In the specific case of instructional quality, the construct might mirror the academic framework and curriculum structure of the broader learning environment [12,13]. It embraces academic traditions and the shared perceptions about teaching from the larger school group about the organisational learning environment, which is also linked to school climate [11,14,62]. Each dimension, distinguished in the construct of instructional quality, can as such be approached from a level going beyond the classroom level.

For example, the dimension of the classroom disciplinary climate of schools refers to the stability and effectiveness of school rules and the frequency of disciplinary incidents [63–65]. The literature provides a compelling basis for predicting a strong connection between school-level disciplinary climate and school performance [63,66,67]. A supportive school climate relates to the availability of organisational academic and emotional supports to

students, resulting in improved learning performance [68–70]. However, inconsistent findings have been found in the literature on whether a supportive school climate has an impact on school performance [62,71]. Regarding the dimension of cognitive activation, studies suggest that cognitive activation would be relatively independent of the school context [72]. Nevertheless, as reflected in TALIS 2013, cognitive activation can also be captured at the school level and considers shared features of teaching methods and problem-based learning [73].

In view of the present study, the former implies that the present study will incorporate a school-level perspective in studying the relationship between instructional quality and mathematics performance in a comparative cross-country setting.

### 2.3. Adding the Focus on School Socioeconomic Status

The importance of students' family socioeconomic status (SES) for learning outcomes is repeated in a univocal way in the literature [74–77]. The concept of SES reflects the position of the individual or family in a hierarchical social structure [78]. Family SES is typically measured by parental education, occupation, and family income [79,80]. Collective SES is the average family SES of all membered students when adopting a multilevel perspective, e.g., classroom or school levels [81]. Collective SES represents socio-demographic characteristics of the neighbourhood school and serves and shapes the overall learning environment through its connection with contextual effects, peer effects, social stratification, educational choice, and institutional differentiation, which may impact educational outcomes [74,82,83]. The strong connection between collective SES and educational performance urges researchers to consider collective SES when analysing instructional quality and academic achievement.

Contextual effects imply that school SES influences academic performance beyond individual socioeconomic background [6,75,84,85]. There is growing interest in assessing the relationship between SES and student achievement when considering the instructional quality [86,87]. Students in low-SES schools receive less effective teaching time compared to students in high-SES schools because of more time-consuming disciplinary incidents [88,89]. Analysis of the PISA 2003 data from 28 OECD countries showed how students in low-SES schools receive more teacher support and finally attain higher mathematics scores [71]. However, research results are still inconclusive on the mediating mechanisms of specific dimensions of instructional quality in the relationship between school SES and academic achievement, especially from a cross-country perspective.

### 2.4. School Characteristics as Predictors of Educational Attainment

School characteristics are manifold. Besides school socioeconomic status, some studies show that schools differ in instructional quality and educational performance regarding school-average qualified teachers (e.g., years of work experience, degree of formal educational background) and teacher–student relationships [6,14,90].

Teachers are one of the vital school-related factors contributing to learning outcomes [6,91]. A growing amount of evidence suggests that certain teacher qualifications do account for part of the differences in teachers' impact on educational attainment [20,22]. Teacher qualifications refer to teachers' verbal and general academic skills and pedagogical content knowledge, such as the degree level of certification in specific subjects, and their years of teaching work experience [20,22]. As organisational characteristics, teachers' collective qualifications vary among schools [92], resulting in an impact on overall teaching behaviours, instructional communication, and learning outcomes [92–95]. Some studies indicate that the employment of qualified teachers in schools is influenced by school-level socioeconomic status [17–19]. For example, a study conducted by Qin and Bowen [96] using TALIS 2013 data across 32 OECD countries found significantly different rates and gaps in exposure to unqualified teachers in schools with low and high SES. Specifically, low-SES schools are more likely have less qualified teachers. However, the inconsistent effect of teacher qualification in different disciplines [97,98], such as reading and mathematics, is

noted. Rare studies have incorporated collective teacher qualification into the mechanism of instructional quality and academic performance regarding school SES.

Teacher–student relationship represents teachers and students going through a process of meeting one another, exchanging information, communicating academic content, adjusting expectations, and achieving goals [26,99]. A positive teacher–student relationship is related to students' emotional, cognitive, and behavioural development [100–102]; furthermore, it significantly influences students' academic performance [103–105]. Even though several studies have highlighted the mechanism between school collective SES and general teacher–student relationships [23–25], there is little empirical knowledge of the extent of the relationship between instructional quality and academic performance in terms of school SES, collective qualifications of teachers, and overall teacher–student relationship.

Given this background, the present article focuses on instructional quality at the school level and its association with mathematics performance, considering school context characteristics. The following two research questions guide the study:

(1) What are the measurement properties of schools' mathematics instructional quality in different countries? Is it appropriate to make a cross-country comparison?
(2) To what extent is mathematics instructional quality related to school mathematics performance in terms of school context features (i.e., school average of socioeconomic status, teacher qualifications, and teacher–student relationship)?

### 3. Methods

#### 3.1. Data and Participants

A linkage dataset of TALIS 2013 and PISA 2012 was used in the present study. Whereas TALIS investigates teachers' learning environments and working conditions, mainly in junior secondary education [106], PISA focuses on capturing the mathematics, reading, and science literacy of 15-year-old students [107]. The primary domain of the PISA 2012 cycle was mathematics. TALIS 2013 also collected data from mathematics teachers who taught 15-year-old students from the same schools participating in the PISA 2012 cycle. A general TALIS 2013–PISA 2012 Link database is available that offers a school-level perspective on student performance [31,108,109]. On the basis of the general linkage database, we developed a specific linkage dataset containing the indicators from students' self-reported data in PISA 2012 (e.g., socioeconomic status, mathematics achievement) and the school profile of mathematics teachers' self-reported data in TALIS 2013 (e.g., teacher work experience, teacher collaboration, teacher self-efficacy). A description and discussion of the specific linking procedure can be found in Liu et al. [29].

From eight countries with data in the linkage database, seven were selected, excluding Mexico (The current study is a follow-up to other studies in which we applied linkage data. In the previous study, the measurement model for Mexico was different from any other country. In view of consistency, Mexico was therefore not included in the analysis in the present study). Table 1 presents the sample sizes per country, resulting in a total of 29,157 students from 1028 schools.

**Table 1.** The sample size of students and schools in each educational system.

|          | Australia | Finland | Latvia | Portugal | Romania | Singapore | Spain |
|----------|-----------|---------|--------|----------|---------|-----------|-------|
| Students | 2205      | 3942    | 1997   | 3832     | 4094    | 5275      | 7812  |
| Schools  | 113       | 133     | 85     | 131      | 131     | 164       | 271   |

#### 3.2. Indicator Selection

In PISA 2012, the SES—Economic, Social, and Cultural Status—is a composite construct derived from the students' self-reported parental education and occupation, family wealth, and home educational and cultural resources. The first plausible value of mathematics achievement, PV1MATH, was used as the outcome variable [110].

School-level indicators for teacher qualification and teacher–student relationship were obtained from the TALIS data. Teacher qualification is an index comprising the highest level of teacher formal education and years of work experience. Teacher–student relationship is an index calculated from four items (i.e., teachers and students usually get on well with each other; most teachers in this school believe that the students' well-being is important; most teachers in this school are interested in what students have to say; if a student from this school needs extra assistance, the school provides it) using a four-point scale, with response categories ranging from 'strongly disagree' to 'strongly agree'. Related descriptive information is reported in Appendix A. For more information about each scale and related items, see TALIS 2013 and PISA 2012 technical reports [73,107].

### 3.3. School-Level Mathematics Instructional Quality Measurement

Twelve items from TALIS 2013 were used to measure the dimensionality of mathematics instructional quality. Four items in TT2G41 are related to the 'learning environment', four items in TT2G42 and TT2G43 describe the 'supportive environment' from teacher perspectives, and four items in TT2M13 refer to 'cognitive activation' in teaching mathematics (see Table 2).

**Table 2.** Item information of mathematics teaching quality from teacher perspectives in TALIS 2013.

| Variable | Item Wording | Response Category |
|---|---|---|
| *Disciplinary climate* | | |
| TT2G41A | I lose quite a lot of time because of students interrupting the lesson | Four-point scale: 1 strongly disagree . . . 4 strongly agree |
| TT2G41B | When the lesson begins, I have to wait quite a long time for students to quiet down | |
| TT2G41C | Students in this class take care to create a pleasant learning atmosphere | |
| TT2G41D | There is much disruptive noise in this classroom | |
| *Support climate* | | |
| TT2G42C | I give different work to the students who have difficulties learning and/or to those who can advance faster | Four-point scale: 1 never or almost never . . . 4 in all or nearly all lessons |
| TT2G43D | I provide written feedback on student work in addition to a mark, i.e., numeric score or letter grade | |
| TT2G43E | I let students evaluate their own progress | |
| TT2G43F | I observe students when working on particular tasks and provide immediate feedback | |
| *Cognitive activation* | | |
| TT2M13C | I expect students to explain their thinking on complex problems | Four-point scale: 1 never or almost never . . . 4 in all or nearly all lessons |
| TT2M13E | I connect mathematics concepts I teach to uses of those concepts outside of school | |
| TT2M13F | I encourage students to solve problems in more than one way | |
| TT2M13G | I require students to provide written explanations of how they solve problems | |

### 3.4. Analysis Approach

All analyses were carried out with Mplus 8.3 [111]. Confirmatory factor analysis (CFA) was performed to test the construct validity of instructional quality. Multilevel Structural Equation Modelling (MSEM) estimated the latent variables—dimensions of schools' mathematics instructional quality—from the observed indicators and helped to model the relationship between socioeconomic status, schools' teaching qualification, instructional quality, and mathematics performance. Maximum likelihood estimation with robust stan-

dard errors (MLR) was applied to handle missing and non-normal data. To evaluate the model fit, the model fit indices with the following cut-off values have been taken into account to accept the model: comparative fit index (CFI) $\geq 0.95$, root mean square error approximation (RMSEA) $\leq 0.05$, and root mean square residual (SRMR) $\leq 0.08$ [112].

The factor structure of school instructional quality using teachers' self-reported data was fitted through CFA separately country by country. Next, measurement invariance helped to test the comparability of these CFA models across countries [113–115]. Three nested models with different degrees of equality constraints were specified when testing measurement invariance. A 'configural invariance model' implies that all countries have the same factor structure. A 'metric invariance model' holds all factor loadings equal across all countries and the identical factor structure. A 'scalar invariance model' sets item intercepts equal on a metric-invariant model. The changes in the goodness-of-fit statistics were examined to evaluate whether the measurement models were invariant across the seven countries. We followed related recommendations proposed by Rutkowski and Svetina [116] for a large number of groups (20 or more); the value changes in CFI ($\Delta$CFI) are not less than $-0.020$, and changes in RMSEA ($\Delta$RMSEA) and changes in SRMR ($\Delta$SRMR) are less than 0.030. When the sample size is less than 300 in each group, Chen [117] suggested specific cut-off criteria: $\Delta$CFI $\leq -0.005$ and $\Delta$RMSEA $\geq 0.010$ or $\Delta$SRMR $\geq 0.025$ between the metric and configural invariance model, and $\Delta$CFI $\geq -0.005$ and $\Delta$RMSEA $\geq 0.010$ or $\Delta$SRMR $\geq 0.005$ between the scalar and metric invariance model. These recommended cut-off values are then used in the following research to investigate the rejection rates for different degrees of invariance within each level and for various levels of invariance.

Multilevel Structural Equation Modelling (MSEM) helped to examine how socioeconomic status, teacher qualification and characteristics, and teaching quality were related to mathematics performance at the student and school levels. In view of students' self-reported socioeconomic status and mathematics achievement, we calculated Intraclass Correlation Coefficients (ICCs) to evaluate if they could be included in subsequent multilevel analyses. The ICCs value of socioeconomic status and mathematics achievement are provided in Table 3.

**Table 3.** ICCs for socioeconomic status and mathematics achievement from students' self-reported data in the PISA2012.

|  | Australia | | Finland | | Latvia | | Portugal | | Romania | | Singapore | | Spain | |
|---|---|---|---|---|---|---|---|---|---|---|---|---|---|---|
|  | ICC1 | ICC2 | ICC1 | ICC2 | ICC1 | ICC2 | ICC1 | ICC2 | ICC1 | ICC2 | ICC1 | ICC2 | ICC1 | ICC2 |
| SES | 0.27 | 0.88 | 0.08 | 0.72 | 0.19 | 0.85 | 0.26 | 0.91 | 0.33 | 0.94 | 0.20 | 0.89 | 0.25 | 0.91 |
| PV1MATH | 0.31 | 0.90 | 0.07 | 0.69 | 0.21 | 0.86 | 0.31 | 0.93 | 0.45 | 0.96 | 0.37 | 0.95 | 0.18 | 0.86 |

*Note.* SES: socioeconomic status, PV1MATH: the first plausible value for mathematics achievement.

Two ICC measures can be distinguished. ICC1 represents the proportion of variance of the outcome variable explained by individuals belonging to different groups [118,119]. In the current analysis, ICC1 captures the variation in SES and mathematics achievement scores, which may be due to the fact that students belong to different schools. The higher ICC1 is, the greater the between-school differences in their students' SES and mathematics achievement. ICC1, in many instances, is used as a measure of school segregation. ICC2, on the other hand, measures the reliability of the aggregated-level variable mean by the proportion of observed total variance in group mean scores occurring at the aggregated level [119,120]. The common guidelines for an acceptable ICC level for multilevel modelling are as follows. A value of ICC1 exceeding 0.05 indicates that a multilevel analysis is essential and meaningful to adjust for hierarchical data structure. A value of ICC2 larger than 0.60 implies the reliable aggregation of the within-group data on the group level [118,119,121,122]. The magnitudes of ICC1 and ICC2 enable researchers to assess how the clustering of individuals in higher-level units affects the observed variation in individual and aggregated scores. Take the ICCs for mathematics performance from Australia as an example. The ICC1 value of 0.31 indicates that 31% of the variance in students' mathematics achievement scores is due to systematic between-school differences.

In contrast, the value of ICC2 is 0.90, indicating that 90% of the observed total variance in the aggregated school mean mathematics score does occur at the school level.

Different values of ICC1 and ICC2 imply that schools differ in their SES and mathematics achievement in each country. It also indicates that there is substantial variation between countries. In the next step, the MSEM approach was specified to model the relationships among schools' mathematics instructional quality, socioeconomic status, teacher–student relationship, and mathematics achievement.

## 4. Results

### 4.1. Measurement Model of Schools' Mathematics Instructional Quality (RQ1)

A CFA model of instructional quality was estimated for each country. For five countries (Finland, Singapore, Portugal, Romania, and Spain), the established three factors of schools' mathematics instructional quality can be identified: disciplinary climate (DC), supportive climate (SC), and cognitive activation (CA). The indicators were significantly related to its latent construct, with a majority of factor loadings exceeding 0.30, with the exception of the loading of TT2M13G in Finland, which equals 0.23. All factor loadings, in general, were higher for the DC factor in all countries than those of the other two factors. A significant correlation between SC and CA factors was found in these five countries, exceeding 0.50. However, only for Singapore, all three factors were significantly related to each other. This might explain the outstanding performance of Singapore in the international large-scale assessments.

The three-factor model structure (i.e., disciplinary climate, supportive climate, and cognitive activation) did not fit data from Australia and Latvia, where the SC factor could not be identified, and the CA factor reflected a different structure. In Latvia, TT2M13E was not significantly related to the CA factor. In Australia, two indicators, TT2G42C and TT2G43E, indicated the CA factor significantly. Higher loadings for the DC factor were observed in these two countries. Meanwhile, the factors DC and CA were significantly intercorrelated.

The model fit indices are summarised in Table 4. For all countries, excellent model fit indices are reported. The factor structure and parameter estimate of schools' mathematics instructional quality are also presented in Table 5 for the three-factor model and Table 6 for the two-factor model in the case of Australia and Latvia.

**Table 4.** Fit statistics of the measurement model for each country.

|  | Australia | Finland | Latvia | Portugal | Romania | Singapore | Spain |
|---|---|---|---|---|---|---|---|
| CFI | 0.95 | 1.00 | 1.00 | 1.00 | 1.00 | 0.99 | 0.99 |
| RMSEA | 0.02 | 0.00 | 0.00 | 0.00 | 0.00 | 0.00 | 0.00 |
| SRMR | 0.08 | 0.06 | 0.03 | 0.06 | 0.07 | 0.05 | 0.04 |
| Chi-square | 52.12 | 51.40 | 8.51 | 52.77 | 52.65 | 54.47 | 56.72 |
| df. | 33 | 51 | 13 | 51 | 51 | 51 | 51 |

Based on the teacher data, the three-factor model structure of schools' mathematics instructional quality did fit well in five countries, while the same structure of the DC factor was established in all seven countries. Measurement invariance was therefore tested to ensure comparability between countries. Tables 7 and 8 summarise these analysis results. The differences in fit indices between the configural, metric, and scalar invariance models were consistently within the cut-off values as defined by Chen [117] (Chen (2007) suggested the cut-off criteria $\Delta CFI \leq -0.005$ and $\Delta RMSEA \geq 0.010$ or $\Delta SRMR \geq 0.025$ between the metric and configural invariance model, and $\Delta CFI \geq -0.005$ and $\Delta RMSEA \geq 0.010$ or $\Delta SRMR \geq 0.005$ between the scalar and metric invariance model). The differences between the three tested levels correspond to the predetermined cut-off values. We can conclude that the scalar invariance holds within the three-factor model and latent construct of disciplinary climate. Therefore, a comparison of association and residual invariance or intercept of mathematics instructional quality can be made across the five countries.



Meanwhile, the school's disciplinary climate can be compared across the seven countries regarding correlations and mean value.

**Table 5.** Three-factor model structures of schools' mathematics teaching quality from teacher perspectives for five countries.

| Scale | Variable | Finland | Portugal | Romania | Singapore | Spain |
|---|---|---|---|---|---|---|
| Disciplinary Climate (DC) | * TT2G41A | 0.86 | 0.80 | 0.79 | 0.73 | 0.79 |
| | TT2G41B | 0.60 | 0.82 | 0.58 | 0.61 | 0.63 |
| | * TT2G41C | 0.88 | 0.95 | 0.91 | 0.90 | 0.87 |
| | * TT2G41D | 0.91 | 0.86 | 0.87 | 0.91 | 0.89 |
| Supportive Climate (SC) | TT2G42C | 0.38 | 0.58 | 0.53 | 0.60 | 0.45 |
| | TT2G43D | 0.50 | 0.37 | 0.45 | 0.57 | 0.44 |
| | TT2G43E | 0.74 | 0.61 | 0.55 | 0.61 | 0.61 |
| | TT2G43F | 0.36 | 0.61 | 0.84 | 0.48 | 0.59 |
| Cognitive Activation (CA) | TT2M13C | 0.60 | 0.68 | 0.62 | 0.56 | 0.46 |
| | TT2M13E | 0.62 | 0.38 | 0.56 | 0.54 | 0.53 |
| | TT2M13F | 0.55 | 0.71 | 0.71 | 0.67 | 0.54 |
| | TT2M13G | 0.23 | 0.77 | 0.55 | 0.36 | 0.51 |
| **Correlation Coefficient** | | | | | | |
| SC with CA | | 0.60 | 0.64 | 0.56 | 0.58 | 0.51 |
| SC with DC | | *0.22* | *0.11* | *−0.18* | 0.28 | *−0.12* |
| CA with DC | | *0.26* | *0.14* | *0.07* | 0.33 | *0.10* |

*Note.* Italic font indicates insignificant estimates. * Item was reverse-coded.

**Table 6.** Two-factor model structures of schools' mathematics teaching quality from teacher perspectives in Australia and Latvia.

| Scale | Variable | Australia | Latvia |
|---|---|---|---|
| Disciplinary Climate (DC) | * TT2G41A | 0.88 | 0.83 |
| | TT2G41B | 0.79 | 0.81 |
| | * TT2G41C | 0.94 | 0.78 |
| | * TT2G41D | 0.88 | 0.81 |
| Cognitive Activation (CA) | TT2G42C | 0.35 | † |
| | TT2G43E | 0.41 | † |
| | TT2M13C | 0.75 | 0.64 |
| | TT2M13E | 0.38 | † |
| | TT2M13F | 0.51 | 0.54 |
| | TT2M13G | 0.57 | 0.77 |
| **Correlation Coefficient** | | | |
| CA with DC | | 0.28 | 0.48 |

*Note.* * Item was reverse-coded. † Insignificant measurable indicator.

**Table 7.** The measurement invariance model fit indices of the three-factor model of schools' mathematics instructional quality across five countries.

| | | CFI | RMSEA | SRMR | $\chi^2$ | df | ΔCFI | ΔRMSEA | ΔSRMR |
|---|---|---|---|---|---|---|---|---|---|
| **Instructional quality** | Configural | 0.983 | 0.005 | 0.058 | 285.112 | 255 | | | |
| | Metric | 0.978 | 0.005 | 0.096 | 332.202 | 291 | −0.005 | 0.000 | 0.038 |
| | Scalar | 0.579 | 0.022 | 0.231 | 1119.342 | 327 | −0.399 | 0.017 | 0.135 |

**Table 8.** The results for the measurement invariance test of disciplinary climate between seven countries.

| | | CFI | RMSEA | SRMR | $\chi^2$ | df | ΔCFI | ΔRMSEA | ΔSRMR |
|---|---|---|---|---|---|---|---|---|---|
| **Disciplinary climate (DC)** | Configural | 1.000 | 0.000 | 0.012 | 10.732 | 16 | | | |
| | Metric | 0.996 | 0.006 | 0.131 | 41.990 | 37 | −0.004 | 0.006 | 0.119 |
| | Scalar | 0.929 | 0.021 | 0.157 | 158.749 | 58 | −0.067 | 0.015 | 0.026 |

### 4.2. Relationship between Mathematics Instructional Quality and School Mathematics Performance (RQ2)

Multilevel Structural Equation Modelling was carried out to investigate the relationships between schools' mathematics instructional quality and mathematics achievement in terms of school-based SES, teacher qualifications, and the teacher–student relationship. Table 9 presents the goodness-of-fit indices of the models in each educational system. Figures 1–7 summarise the associations. In the path diagrams of Figures 1–7, the observed variables are represented in boxes, and the latent factors are represented in circles. Significant relationships between factors are depicted with single arrow lines, with the head of the arrow pointing towards the variable being influenced by another factor in the current study. The path coefficients specify the relationships among the specified factors. The curved double arrow lines imply two latent factors being correlated. The residual variance or the measurement errors are the values represented next to the factor with single arrow lines.

**Table 9.** The model fit indices of the two-level model analysis.

|  | **Australia** | **Finland** | **Latvia** | **Portugal** | **Romania** | **Singapore** | **Spain** |
|---|---|---|---|---|---|---|---|
| CFI | 0.95 | 0.97 | 1.00 | 0.98 | 0.97 | 0.98 | 0.99 |
| RMSEA | 0.02 | 0.01 | 0.00 | 0.01 | 0.01 | 0.01 | 0.00 |
| SRMR-within | 0.00 | 0.00 | 0.00 | 0.00 | 0.00 | 0.00 | 0.00 |
| SRMR-between | 0.08 | 0.07 | 0.04 | 0.06 | 0.07 | 0.06 | 0.04 |
| Chi-square | 95.39 | 108.92 | 27.40 | 102.69 | 107.38 | 108.20 | 98.68 |
| df. | 65 | 87 | 33 | 87 | 87 | 87 | 87 |

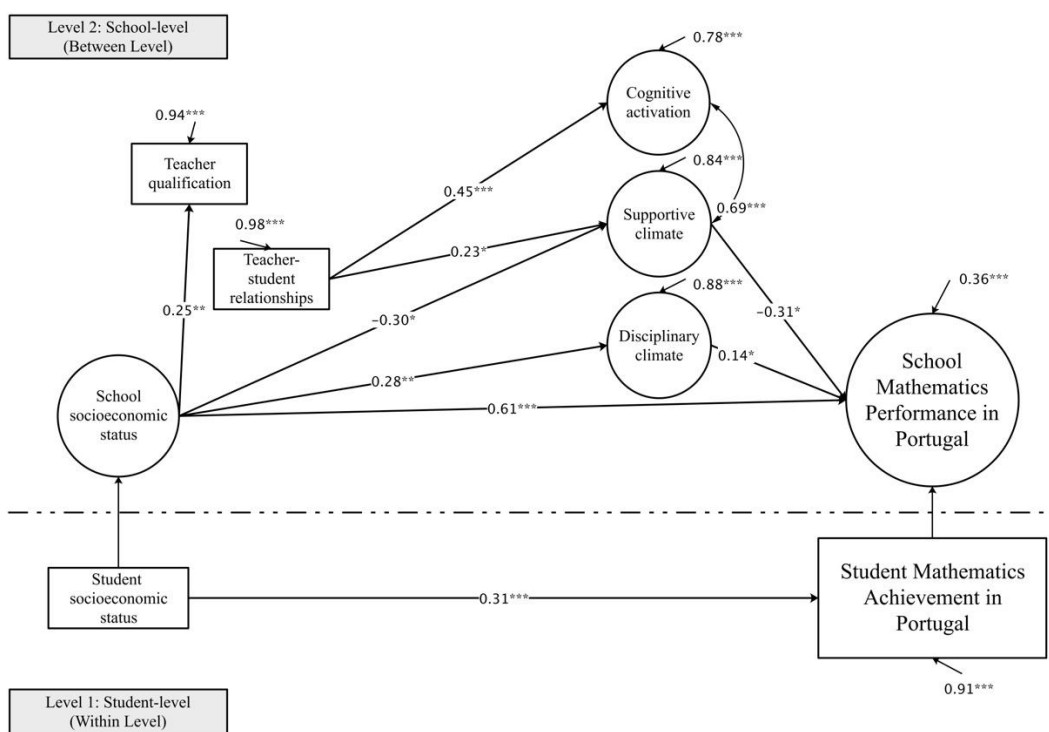

**Figure 1.** Path diagram for Portugal. *Note. Only significant paths are shown (\*\*\* p < 0.001, \*\* p < 0.01, \* p < 0.05).*

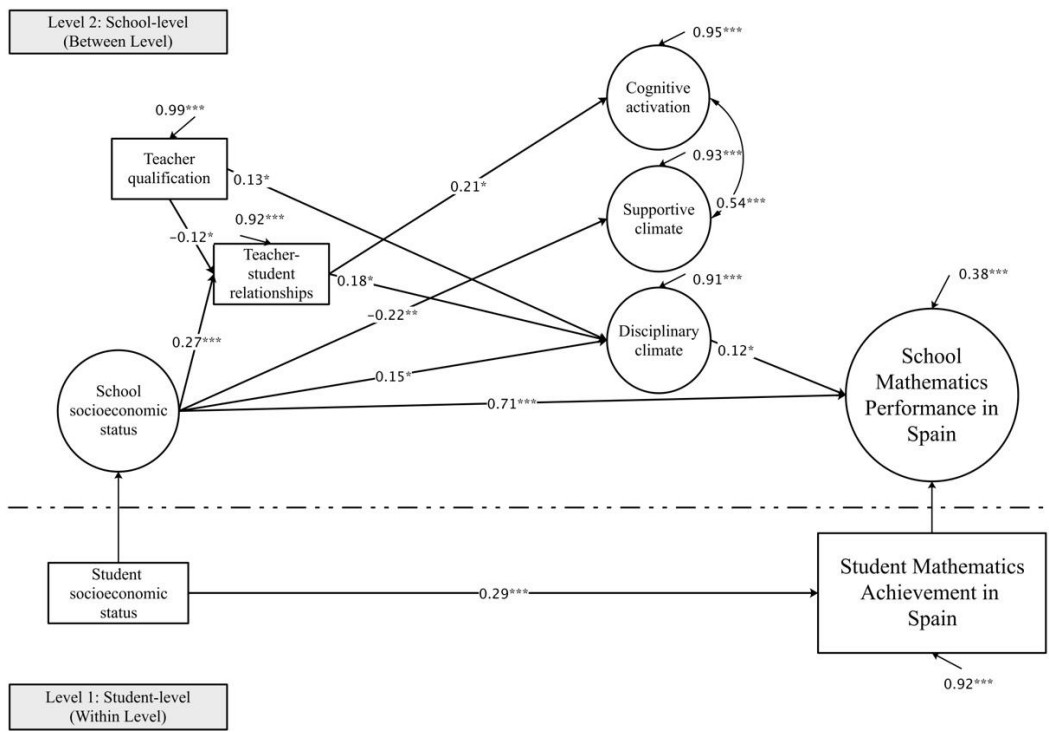

**Figure 2.** Path diagram for Spain. *Note. Only significant paths are shown* (*** *p* < 0.001, ** *p* < 0.01, * *p* < 0.05).

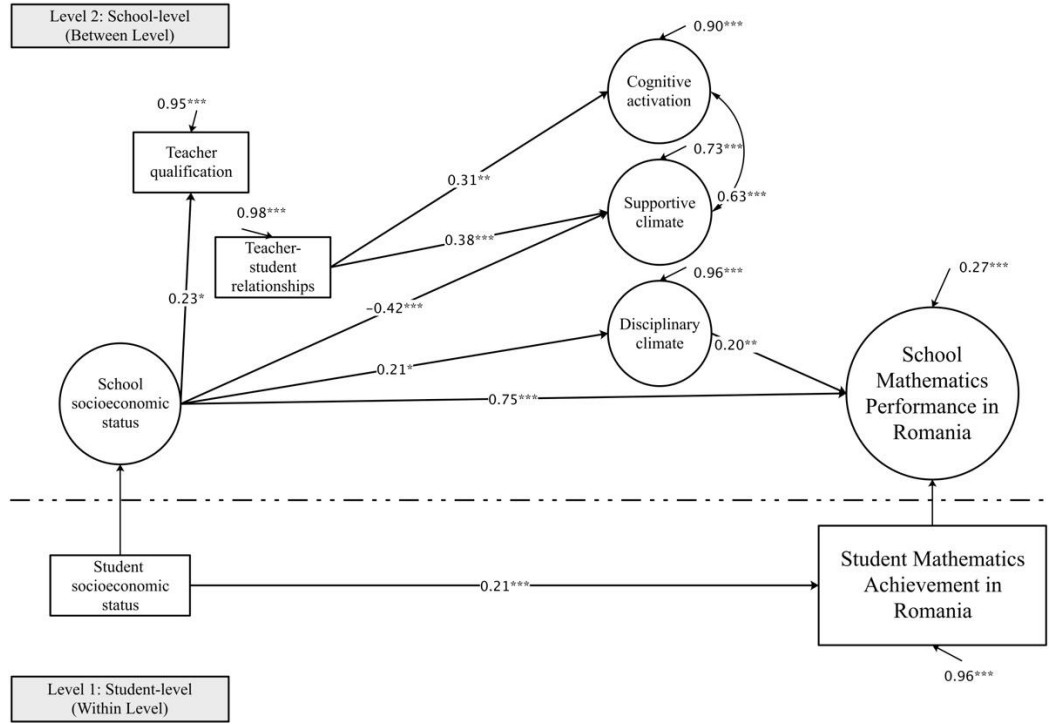

**Figure 3.** Path diagram for Romania. *Note. Only significant paths are shown* (*** *p* < 0.001, ** *p* < 0.01, * *p* < 0.05).

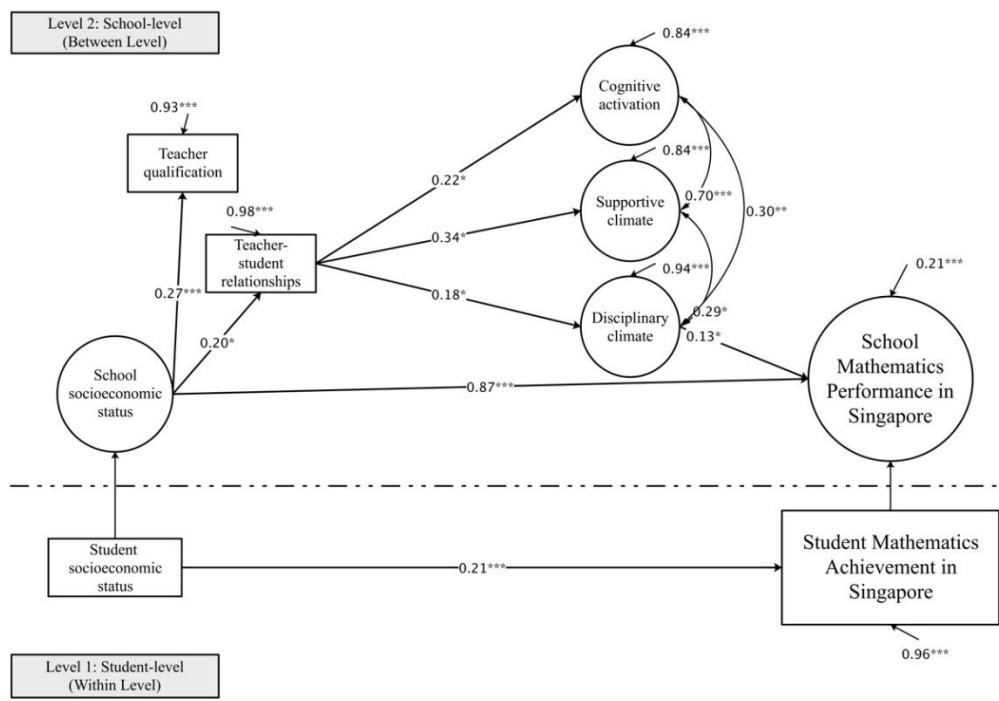

**Figure 4.** Path diagram for Singapore. *Note. Only significant paths are shown* (*** *p* < 0.001, ** *p* < 0.01, * *p* < 0.05).

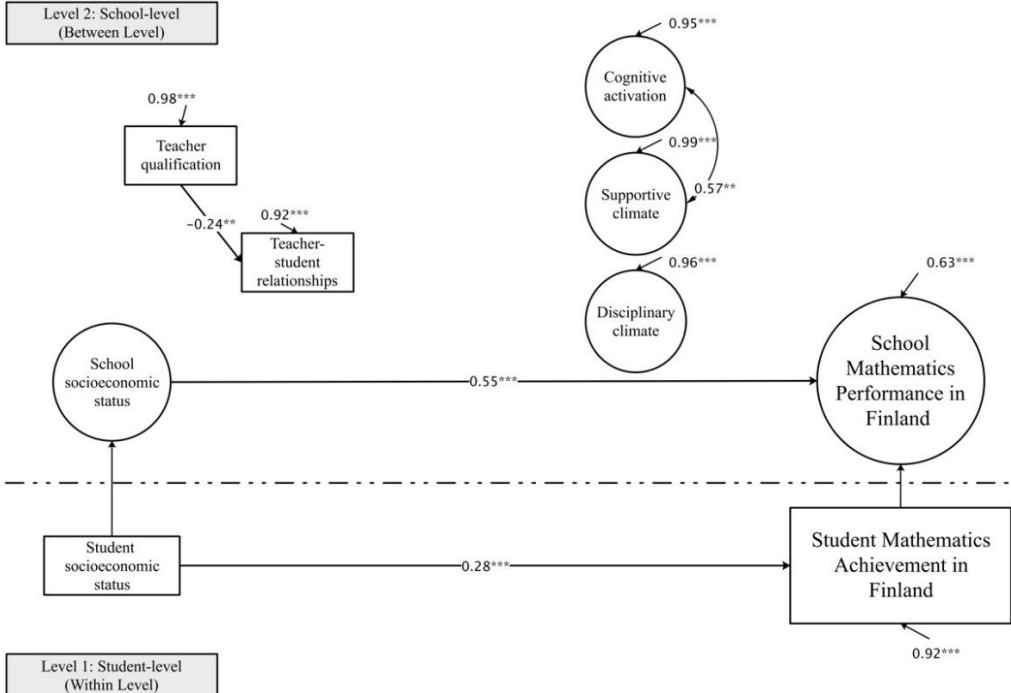

**Figure 5.** Path diagram for Finland. *Note. Only significant paths are shown* (*** *p* < 0.001, ** *p* < 0.01).

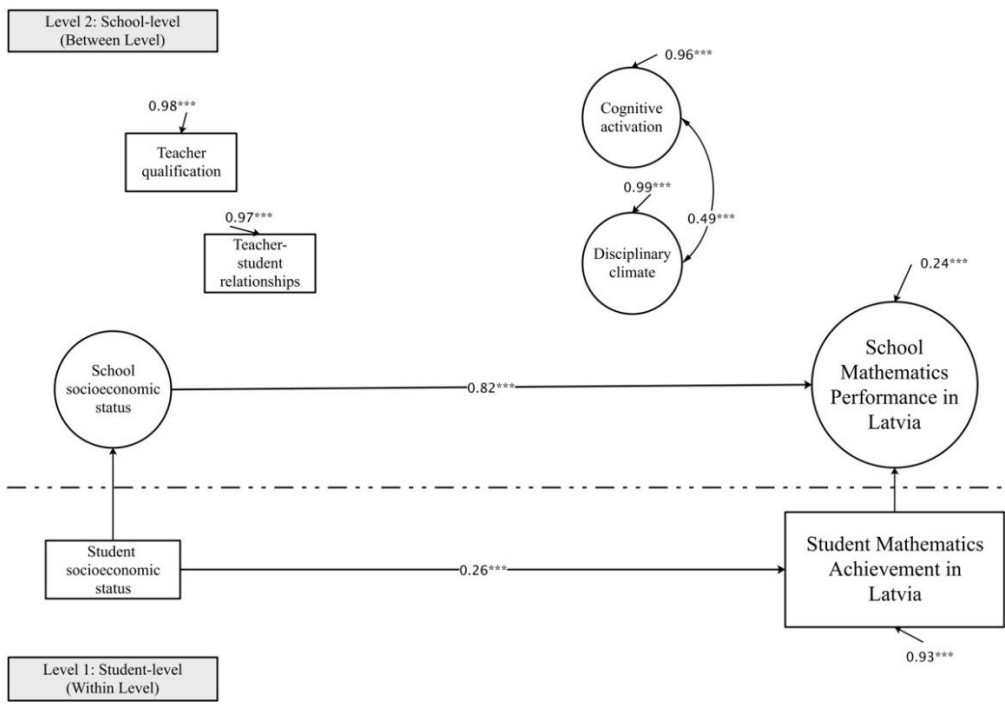

**Figure 6.** Path diagram for Latvia. *Note. Only significant paths are shown (*** p < 0.001).*

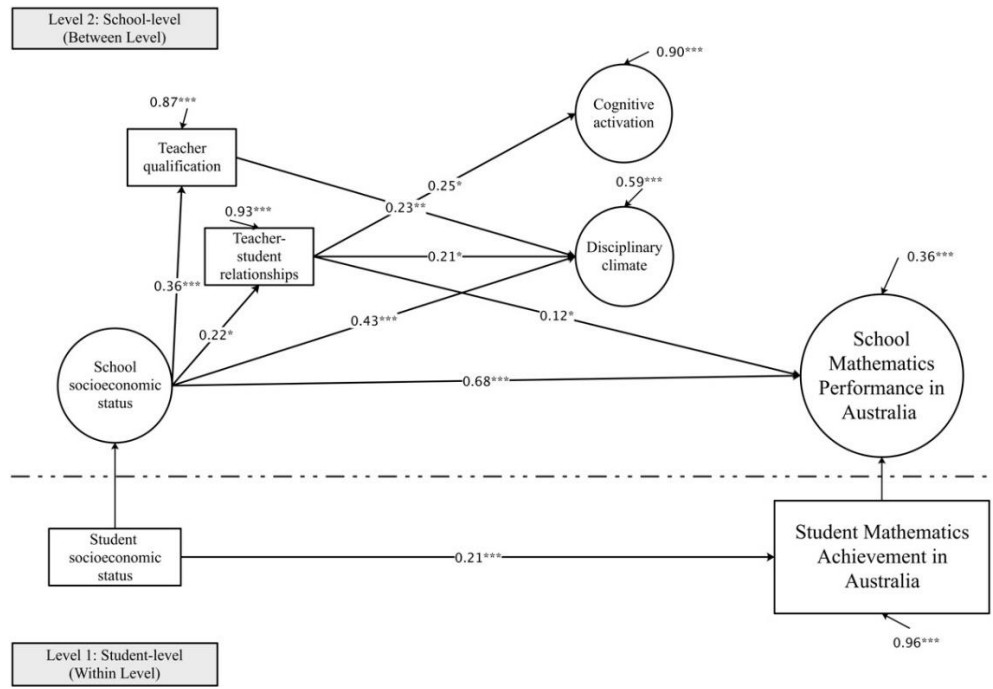

**Figure 7.** Path diagram for Australia. *Note. Only significant paths are shown (*** p < 0.001, ** p < 0.01, * p < 0.05).*

The mechanism patterns vary across educational systems. The effects of SES on mathematics performance were significant at the student and school levels. Consistent with the previous analysis, school SES had a greater effect than student family SES, which was also the strongest predictor of school mathematics performance (from 0.55 to 0.87).

Four countries where a three-factor model structure (i.e., disciplinary climate, supportive climate, and cognitive activation) for instructional quality in school mathematics could be confirmed (i.e., Portugal, Spain, Romania, and Singapore; see Figures 1–4), and where

the factor structure for the disciplinary climate was shared (from 0.12 to 0.20), also showed positive connections to school mathematics performance. In the three European countries, schools with predominantly students from favourable socioeconomic families (school SES) reflect a stronger disciplinary climate, which is more conducive to learning and results in better performance. In Spain and Singapore, schools with a stronger disciplined climate mirror higher performance since, in these schools, we observe more students from an advantaged SES background and more harmonious teacher–student relationships. Although school collective teacher qualification—as an organisational characteristic—seems not directly related to school performance, disciplinary climate mediates this link in Spain.

The factor of supportive climate is only in Portugal a significant and negative predictor. Teacher–student relationship has a positive effect on supportive climate. The results demonstrate that positive teacher–student relationships contribute to a favourable helping environment, which in turn provides more help in low-performing schools.

Despite the fact that mathematics instructional quality in Finnish schools also reflects three dimensions, none is related to school mathematics performance, and we observe how the impact of school SES is the smallest among all seven countries. A plausible explanation is that the Finish educational policy about equality has contributed to the small differences between Finnish schools. Simultaneously, studies also point at the classroom-level explanations for the related variation in mathematics performance [123]. We discuss the findings of Finland in more detail in the next section.

In Australia and Latvia, two countries reflecting a two-factor instructional quality model, none of the dimensions of mathematics instructional quality predicts school performance (see Figures 6 and 7). However, the direct and mediated relationship between teacher–student relationship and school performance is observed in Australia. This implies that schools with harmonious teacher–student relationships may have differential effects on mathematics performance in schools with students from lower- and higher-socioeconomic status families.

## 5. Discussion

The central aim of the present study was to investigate the measurement properties and mechanism patterns between mathematics instructional quality and school performance regarding school context characteristics (i.e., collective socioeconomic status, teacher qualification, and teacher–student relationship). The comparative perspective across seven countries provides insights into diverse cultures and reflects the activities of the school systems in teaching mathematics. The first application of TALIS 2013 and PISA 2012 linked data focused on the 'mathematics domain', which sheds novel light on the interpretation of mathematics performance. The current findings underline the importance of understanding socioeconomic status for academic performance and highlight the role of two dimensions of mathematics instructional quality (i.e., disciplinary climate and supportive climate) in school mathematics performance. Our results have several theoretical and practical implications that contribute to the study of mathematics instructional quality at the school level, as follows.

### 5.1. Multi-Dimensional Nature of School's Mathematics Instructional Quality

The present study adopted a rather different stance in studying instructional quality for a number of reasons. The first reason is related to the nature of the data: self-reported data from TALIS 2013 were used to study mathematics teaching at the classroom and school levels. The results are largely in line with previous studies about the subject-specific impact of instructional quality [32,33,124,125]. This reiterates the importance of considering domain-specific knowledge and domain characteristics when studying instructional quality in the math domain. The authors point to the specific mathematical language and mathematical concepts that induce such differences [33].

Building on teachers' self-reported data, earlier studies stressed mathematics instructional quality dimensions, such as cognitive activation, classroom management, supportive

climate, classroom disciplinary climate, assessment and feedback, and clarity of instruction [9,39,49,53]. However, to date, most related studies have analysed this at the classroom level and less in terms of the school-level teaching quality of schools [53,60,126]. Instructional quality in school effectiveness research introduces such focus at the school level. When transferring the focus from the classroom/teacher level to a school-level setting, we expect differences in conceptualisation between individuals and groups [61,127,128].

For example, in some countries, the measurement properties of supportive climate differ at the teacher level from the school level. At the teacher level, this dimension consists of four indicators in Australia. However, at the school level, supportive climate cannot be identified in Australia, or in Latvia. In TALIS 2013, four indicators are used to describe this dimension by referring to teachers' attentive and sensitive attempts to resolve understanding difficulties in the classroom (see Table 2). At the school level, these indicators could refer to general support mechanisms for academic learning and shared teaching approaches. Reasons that this might not be identified in, e.g., Australia and Latvia, can be due to countries' systems and school composition differences. First, the federal educational authorities largely influence schools in Australia, implying that schools have no choice in educational policy, teacher training content, and teaching methods [129]. When looking at Latvia, the educational system has gradually moved away from a centralist Soviet system since 1991, but despite these changes, the current Ministry of Education and Science [130] remains the main policy decision-making body, responsible for setting educational standards, teaching content, and pedagogical processes [131,132]. Secondly, according to the Population Census 2011, populations in Australia [133] and Latvia [134] are characterised by a large number of diverse cultural and multilingual ethnic groups. This might lead to weaknesses in shaping uniform educational standards that do not fit the academic needs of students with a multicultural background. The former leads to our finding that a school-level supportive climate is not identified in Australia and Latvia.

Previous research suggests that choosing the appropriate level of analysis depends on the research questions [61,127]. The current study addresses how instructional quality based on teacher perception data is related to mathematics performance at the school level. This introduced the need to re-assess the construct of mathematics instructional quality at the school level. Therefore, the present study contributes to our understanding of the dimensions of schools' mathematics instructional quality from a teacher perspective and how these factors influence school-level mathematics performance.

### 5.2. The Role of Disciplinary Climate

Another aim of the current study was to test how instructional quality dimensions help to predict school performance in mathematics. As a common factor, disciplinary climate helped to explain the variation in school mathematics achievement in four out of seven countries (i.e., Portugal, Spain, Romania, and Singapore). These countries can be compared in terms of shared or different cultural values. The Asian country—Singapore—can be looked at through the lens of a Confucian culture in which the teacher establishes the line of authority and clarifies students who are in charge of the learning environment [63,67]. The latter implies that a strict disciplinary climate is expected to ensure the successful transmission of knowledge. Looking at Romania, an Eastern European country, it reflects an emphasis on hierarchy and obeyance to authority [135]; this could explain the willingness of students to follow orders from superiors (e.g., teachers) and respect a disciplinary climate. This helps to understand why the disciplinary climate has a stronger impact in Romania compared to other countries. Compared to two Southern European countries, Portugal, and Spain, we also observe cultural differences. Spain reflects an intermediate level of collectivism (i.e., the degree to which individuals express pride, loyalty, and cohesiveness in their organisations or families) and individualism (i.e., inducing personal behaviour that is responsible for individual interests) [135–137]. In comparison, Portugal is said to mirror a collectivist and hierarchical culture [135]. Society may favour hierarchical roles to teach the citizens to obey authority, as in Romania. Meanwhile, partly as in Spain, individuals

express loyalty and cohesiveness to local organisations. These country characteristics might help to explain how a school's disciplinary climate plays a role in Spain and Portugal.

The observed positive and significant effect of disciplinary climate is in line with previous research [63,65,67,138]. Nevertheless, most of these studies conceptualise disciplinary climate based on student data and describe disciplinary climate mostly at the classroom level, independent of the school environment. They also do not consider instructional quality as a key construct [65,139,140]. Our findings enrich the study of disciplinary climate from a teacher's perspective and extends, e.g., earlier studies—such as that by Liu, Yang Hansen [52]—that primarily identified classroom disciplinary climate as a dimension at the teacher level. For the first time, our study used TALIS teacher data to define schools' overall mathematics instructional quality and documented the associations with school performance. In line with the acknowledged mediating role of disciplinary climate in understanding learning outcomes [63,139], our findings add the mediation mechanisms of school disciplinary climate between socioeconomic status, teacher–student relationships, and mathematics performance, while looking from a school-level perspective.

The reasonable explanation for the above is that schools composed of students with a high-SES background tend to perform well because of a shared cultural identity between their home and the school [141]. Groups of students with similar SES backgrounds typically exhibit similar communication habits, moral standards, behavioural norms, and interpersonal relationships, thus achieving similar academic results [25,142,143]. A harmonious teacher–student relationship—in this context—is beneficial in overcoming minor disciplinary incidents to form a positive disciplinary climate [144], which indirectly and directly affects academic performance [27,28].

Although the mediating mechanisms of disciplinary climate between school-level collective teacher qualifications (i.e., educational background, working experiences) and school performance are only significant in Spain, the findings further enrich the potential mediating role of school disciplinary climate. Collective teacher qualification refers to the shared characteristics of teachers in schools, such as general subject knowledge, pedagogical content knowledge in mathematics education, verbal skills, and teaching motivation [20,22]. Schools hiring and attracting more qualified teachers may adapt the teaching behaviours, enhance the factor of instructional quality, and make a difference in academic student performance [21,145–147]. Our findings are consistent with the previous model regarding teaching factors as inputs to teaching behaviours and instructional quality [21,45,148]. While our study found an association between the school profile of SES and teacher qualification, high-SES schools tend to hire more academically qualified teachers. We could also hypothesise that highly qualified teachers prefer working in schools with a favourable SES composition. However, we have no evidence to support whether this relationship affects learning outcomes.

### 5.3. Towards a Country-Specific Model or a Universal Model for Instructional Quality

Cross-cultural perspectives help us to understand the conception and dimensional structure of mathematics instructional quality, the interdependency among teacher qualification, socioeconomic status, and teacher–student relationships, and how this relates to mathematics performance in the school context. The selected seven countries in the current study are from different geographical locations and mirror diverse cultural characteristics, representing different education systems in Asia and Europe. We explored the factor structure of mathematics instructional quality more closely in each system and found that the county comparison reached a scalar invariance level. Even though some countries seem to reflect a comparable factor structure when looking at school-level instructional quality in mathematics education, the findings nevertheless reflect country-specific mechanisms and patterns. The effects of instructional quality on school mathematics performance differ across countries since the educational context influences the implementation of mathematics teacher education [149]. Teacher knowledge and skills depend on, e.g., the country-specific duration and level of teacher education, specific pre- and in-service

training programmes, and other professional development activities. This leads to overlap and differences in the quality of instruction in different countries [150].

There is consistency in the results from the seven countries when looking at students from advantaged SES families that perform better in mathematics achievement and that high-SES schools also reflect better school mathematics performance. School-average SES has a strong impact on achievement, which is higher than the impact of family SES in all countries. In none of the countries could we identify the dimension of cognitive activation. This seems in accordance with our hypothesis and previous studies wherein cognitive activation is independent of the school environment and plays a role at the classroom level [72]. Although the relationship between school SES and teacher qualifications was found in some countries (i.e., Australia, Portugal, Romania, and Singapore), teacher qualifications did not affect achievement. Nevertheless, it is important to repeat that schools with high SES attract more competent teachers, resulting in the unbalanced distribution of qualified teachers among academically and economically advantaged schools and students.

The results from Finland are worth discussing since school SES only explains a relatively small proportion of the variation in school mathematics achievement compared to other countries. It was the only country where all three dimensions of mathematics instructional quality could be identified, but also where none was related to school performance. At the same time, other school context features were not related to mathematics instructional quality and achievement. Only 7% of the variation in school mathematics performance could be related to the education system in Finland. An explanation could be that Finland is one of the most equitable and top-performing countries globally, with minor differences between schools and a high level of student inclusion [151,152]. The decentralised education system and highly competent teachers might lead to more autonomy and higher quality of teaching [152,153]. Substantial classroom differences and large classroom variations in performance further help to explain why Finnish instructional quality at the school level is not related to achievement [45,123].

Another country worth discussing is Portugal, where three predictors (i.e., socioeconomic status, disciplinary climate, and supportive climate) explain a large proportion of the variation in school performance in mathematics. This could be attributed to the democratic and decentralised education system, particularly the freedom of schools to decide about ways to teach and learn [154]. Schools enjoy autonomy in terms of school pedagogy, managing teaching schedules and administration and management agencies, as well as the possibility of sharing pedagogical projects with other schools. Based on what was discussed earlier about the hierarchical and collectivist cultural values, the information about the educational system in Portugal helps to explain why a supportive climate is only found in Portugal. Firstly, teachers enjoy the autonomy in choosing the content and the help they provide to students. Secondly, it can be hypothesised that the high cohesion in schools and strong loyalty of the teachers contribute to the willingness of teaching staff to match effective support to the needs of students. Students of lower-performing schools receive more help in Portugal. Meanwhile, a supportive climate reduces the social achievement gap, as schools create a learning environment that shows patience and understanding for students experiencing difficulties, potentially reducing the disadvantage of students from lower socioeconomic backgrounds. This implies that school effectiveness in terms of equity can be evaluated by looking at the extent to which differences in academic performance between groups of students from different backgrounds are being reduced.

### 5.4. Strengths and Limitations

A merit of our study—from a methodological viewpoint—is the adoption of a specific design, i.e., exploring the measurement model and testing measurement invariance. To our knowledge, this is the first attempt to study the effects of mathematics instructional quality and school context features on school mathematics performance using TALIS and PISA linking data. Additionally, the cross-country comparative analysis provided fresh insights into the interplay between country differences and questioned the often taken-for-granted

cultural value, school systems, teaching behaviours, and interpersonal interactions. We also expanded and complemented previous instructional quality research by focusing on the school level and by building on mathematics teacher data. This enriches our understanding of the associations between mathematics instructional quality and school characteristics and how this explains variation in school mathematics performance. Meanwhile, the current study puts the spotlight on the mediated role of disciplinary climate and supportive climate. In particular, schools composed of students with low socioeconomic backgrounds seem to benefit more from a supportive climate, potentially reducing the achievement gap.

However, our study cannot establish evidence by making causal inferences from the TALIS–PISA linkage data. We repeat that the results are based on correlations between predictors and outcome variables. Moreover, despite the switch from studying teacher instead of student data, the former have also been obtained via self-reports. The fact that we analysed data from students and teachers in the mathematics domain also limits possible generalisations. Though we build on large-scale studies, the data from some countries were collected from smaller sample sizes (e.g., Latvia). The former implies that we have underestimated the nature and strengths of relationships between variables.

## 6. Conclusions and Future Outlook

Drawing from a dynamic and hierarchical framework of educational effectiveness, the present study introduced innovations in studying instructional quality and using the TALIS–PISA linkage data. The key was the shift to a focus on the school level and to build on teacher data. This helped to develop and examine a multilevel model about the associations between school context features, school instructional quality, and mathematics achievement. Our study contributes to school effectiveness research by considering the school profile in mathematics instructional quality, combining diverse facets of school characteristics that help to build a more comprehensive model to explain the variation in school mathematics performance. Moreover, understanding instructional quality from mathematics teacher perceptions helps to emphasise the importance of domain-specific knowledge characteristics. The findings have a range of implications for underlining school instructional quality in mathematics education, highlighting to school leadership and education policymakers that strengthening and creating an orderly and supportive climate could be fruitful.

Taken collectively, schools differ systematically in their instructional quality and mathematics performance. Our results reveal a three-dimensional framework of schools' mathematics instructional quality (i.e., disciplinary climate, supportive climate, and cognitive activation) in five countries and a two-dimensional model in two countries (i.e., disciplinary climate and cognitive activation). Based on the results of measurement invariance testing, a three-dimensional framework across seven countries and the dimensions of disciplinary climate between five countries reached the scalar invariance level. Different mechanisms were observed to play a role when comparing countries. Disciplinary climate could explain the variations in mathematics performance and mediate the relationship between school socioeconomic status and mathematics achievement in some countries. Schools composed of students with low socioeconomic backgrounds benefited more from a supportive climate, contributing to the reduction in the achievement gap in Portugal. Schools with harmonious teacher–student relationships may have differential effects on the mathematics performance of school composition with students from lower- and higher-socioeconomic-status families in Australia. In addition, school-level SES is the strongest predictor of academic performance.

The present research suggests that cross-cultural and subject-specific perspectives need to be considered when measuring teaching quality and setting up educational effectiveness research. It would be very informative to the policymakers and international audience to understand the concept of instructional quality and the reasons behind the effects on school performance in other cultural contexts. In future research, we suggest that multi-dimensional instructional quality needs to be supported empirically in different contexts.

It is essential to differentiate the factor structure and conceptualise the dimensions of instructional quality at the group level. The linkage data also provide indicators about the school profile of teacher characteristics (e.g., school beliefs, professional development) and school climate and school culture. Future studies should examine how these factors explain the differences in mathematics performance.

**Author Contributions:** Conceptualization, X.L., M.V. and K.Y.H.; methodology, X.L. and K.Y.H.; software, X.L.; validation, X.L. and K.Y.H.; formal analysis, X.L.; investigation, X.L. and K.Y.H.; resources, X.L.; data curation, X.L.; writing—original draft preparation, X.L.; writing—review and editing, X.L., M.V., K.Y.H. and J.D.N.; visualization, X.L.; supervision, M.V., K.Y.H. and J.D.N.; project administration, M.V.; funding acquisition, X.L. All authors have read and agreed to the published version of the manuscript.

**Funding:** This research was fully funded by the China Scholarship Council (CSC), grant number CSC201807930019, and partially funded by the Fonds Wetenschappelijk Onderzoek (FWO), grant number V412020N.

**Institutional Review Board Statement:** Not applicable.

**Informed Consent Statement:** Not applicable.

**Data Availability Statement:** The study presents secondary public data provided by OECD. The data and materials were retrieved from the TALIS website from https://stats.oecd.org/index.aspx?datasetcode=talis_2013%20 (accessed on 31 March 2022) and the PISA website from https://www.oecd.org/pisa/pisaproducts/pisa2012database-downloadabledata.htm (accessed on 31 March 2022).

**Conflicts of Interest:** The authors declare no conflict of interest.

## Appendix A

**Table A1.** Descriptive statistics of the indicators broken down by country.

|  | Australia | | Finland | | Latvia | | Portugal | | Romania | | Singapore | | Spain | |
|---|---|---|---|---|---|---|---|---|---|---|---|---|---|---|
|  | Mean | Std. Deviation | Mean | Std. Deviation | Mean | Std. Deviation | Mean | Std. Deviation | Mean | Std. Deviation | Mean | Std. Deviation | Mean | Std. Deviation |
| TT2G41A | 2.80 | 0.56 | 2.64 | 0.59 | 2.93 | 0.56 | 2.64 | 0.52 | 3.24 | 0.47 | 2.88 | 0.41 | 2.72 | 0.62 |
| TT2G41B | 2.64 | 0.56 | 2.48 | 0.60 | 2.71 | 0.50 | 2.67 | 0.48 | 2.90 | 0.45 | 2.72 | 0.37 | 2.62 | 0.49 |
| TT2G41C | 2.73 | 0.67 | 2.71 | 0.62 | 3.08 | 0.56 | 2.58 | 0.55 | 3.19 | 0.51 | 2.91 | 0.42 | 2.69 | 0.60 |
| TT2G41D | 2.83 | 0.59 | 2.68 | 0.59 | 2.85 | 0.66 | 2.86 | 0.53 | 3.23 | 0.47 | 2.96 | 0.40 | 2.82 | 0.58 |
| TT2G42C | 2.51 | 0.52 | 2.69 | 0.56 | 2.54 | 0.49 | 2.40 | 0.50 | 2.55 | 0.41 | 2.11 | 0.34 | 2.14 | 0.63 |
| TT2G43D | 2.66 | 0.39 | 1.83 | 0.50 | 1.97 | 0.54 | 2.69 | 0.45 | 2.12 | 0.51 | 2.86 | 0.40 | 2.62 | 0.57 |
| TT2G43E | 2.09 | 0.50 | 1.97 | 0.47 | 2.36 | 0.42 | 2.43 | 0.41 | 2.29 | 0.37 | 2.22 | 0.36 | 1.75 | 0.53 |
| TT2G43F | 3.26 | 0.48 | 2.90 | 0.62 | 3.01 | 0.48 | 3.06 | 0.45 | 2.95 | 0.49 | 3.00 | 0.31 | 3.12 | 0.53 |
| TT2M13C | 2.98 | 0.55 | 2.95 | 0.51 | 2.99 | 0.43 | 3.40 | 0.36 | 2.98 | 0.53 | 2.71 | 0.39 | 2.71 | 0.54 |
| TT2M13E | 2.96 | 0.48 | 2.56 | 0.44 | 2.87 | 0.53 | 3.13 | 0.40 | 2.84 | 0.42 | 2.63 | 0.35 | 2.89 | 0.49 |
| TT2M13F | 2.93 | 0.47 | 2.48 | 0.56 | 3.02 | 0.49 | 3.19 | 0.38 | 3.15 | 0.44 | 2.94 | 0.32 | 3.06 | 0.50 |
| TT2M13G | 2.67 | 0.63 | 3.14 | 0.76 | 2.89 | 0.57 | 3.17 | 0.41 | 2.81 | 0.53 | 2.44 | 0.54 | 2.69 | 0.69 |
| SES | 0.18 | 0.82 | 0.31 | 0.81 | −0.11 | 0.84 | −0.53 | 1.15 | −0.45 | 0.90 | −0.31 | 0.91 | -0.17 | 1.02 |
| EDUBAK | 3.00 | 0.07 | 3.00 | 0.18 | 3.00 | 0.04 | 3.11 | 0.21 | 2.98 | 0.16 | 2.99 | 0.08 | 2.93 | 0.32 |
| WOKEXPT | 7.35 | 4.48 | 12.17 | 6.97 | 16.64 | 7.83 | 12.24 | 5.63 | 15.58 | 6.52 | 7.75 | 3.37 | 11.87 | 7.31 |
| TSCTSTUDS | 13.64 | 1.60 | 13.42 | 1.21 | 12.58 | 1.01 | 13.32 | 1.10 | 12.40 | 1.14 | 12.88 | 0.79 | 12.95 | 1.52 |
| PV1MATH | 495.99 | 99.63 | 511.58 | 86.69 | 501.53 | 79.95 | 484.56 | 92.72 | 449.40 | 79.36 | 569.24 | 105.22 | 488.92 | 88.75 |

*Note.* SES: socioeconomic status; EDUBAK: teacher educational level; WOKEXPT: years of work experience; TSCTSTUDS: teacher–student relationship; PV1MATH: the first plausible value of mathematics achievement.

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
