# Peer review of "Does School-Level Instructional Quality Matter for School Mathematics Performance? Comparing Teacher Data across Seven Countries"

_sustainability, doi:10.3390/su14095267_

Round 1

Reviewer 1 Report

In the presented article, the authors want to establish a more nuanced view of the relationship between the mathematics teachers’ instructional quality and school mathematics performance.
The methods chosen by the authors correspond to the goal. The article represents a holistic study. The conclusions of the article correspond to the content. The text is of scientific and practical pedagogical interest.
However, line 63 starts with "To achieve this". This construction needs an explanation.

Author Response

Thank you for these appreciative comments.

For your suggestion," line 63 starts with "To achieve this". This construction needs an explanation", and we are aware that the expression “To achieve this” is not clear. The description in this paragraph followed the research gaps and aims of the study, we set out in the previous paragraph. 

“Therefore, in the current study, we want to establish a more nuanced view of the relationship between mathematics teachers’ instructional quality and school mathematics performance. We first explore the factor structure of school mathematics instructional quality building on TALIS 2013 mathematics teachers’ data from seven countries[1] (Australia, Finland, Latvia, Portugal, Romania, Spain, and Singapore).”

[1] When implementing TALIS 2013, participating countries could opt for an extra PISA-related extra survey that required all mathematics teachers to participate in the Mathematics Teacher Questionnaire. This extra option is labeled as the TALIS-PISA Link. More specifically, mathematics teachers were considered to be a subsample of teachers who did teach students in schools that participated in PISA 2012. Eight countries opted to participate in this extra survey: Australia, Finland, Latvia, Mexico, Portugal, Romania, Singapore, and Spain. The current study is a follow-up to other studies in which we applied linkage data. In the previous study, the measurement model for Mexico was different from any other country. In view of consistency, Mexico was therefore not included in the analysis in the present study. Thus, seven countries were selected in the following analysis.

Reviewer 2 Report

The research problem posed, on the one hand the relationship between instructional quality and students assessment and on the other hand the comparison between different countries is a topic of special interest in mathematics education. The data used, coming from international tests, are free of biases of any kind.

The article identifies the variables or dimensions of study and they are clearly described.

The data, coming from international reports such as TALIS and PISA, are adequate as they allow relating in each country (and even in each school) the variables on teachers with the variables on students. They are also data that are homogeneous across countries and are therefore relevant for the comparative study between different countries that have participated in both studies.

The qualitative (statistical) analyses used as an analytical procedure are adequate and tested with coefficients that give plausibility to the results. Statistical methods make it possible to infer which variables are the main variables explaining the phenomenon under study and the relationships between these variables with certain degrees of validity (with specific coefficients), this work explains this aspect of qualitative research in education and gives "reliability" to the results obtained.

Both the quantitative tables and the figures help to understand the results of the article and have a relevant explanatory role.

There is an error in table 9, the last column is labelled as "ESP" which is Spain (in Spanish language) but in English it is Spain.

Author Response

Thank you for these detailed comments and the related appreciation.

For this suggestion, "There is an error in table 9, the last column is labelled as "ESP" which is Spain (in Spanish language) but in English it is Spain", thank you for pointing this out. To avoid such errors in country abbreviations, we now replaced all country abbreviations with full names. This also applies to texts, tables, and figures. 

Please see the new version of the manuscript.

Reviewer 3 Report

The paper should explain the reason why the seven countries were chosen. The are 5 EU and 2 other continent countries, with differenct educational system, differenct a social background.  The expalanation comes much later, and contain some generalities. It could be a reason of chosniug this countries e.eg their result obtained in IMO or other mathematics competrions.  The figures 1-7 are very interesting but their contruction is not explained sufficiently.

Author Response

Thank you for your observation.

For point 1, "The paper should explain the reason why the seven countries were chosen. The are 5 EU and 2 other continent countries, with differenct educational system, different social background. The explanation comes much later, and contains some generalities. It could be a reason of chosing these countries e.g. their result obtained in IMO or other mathematics competrions", we restructured the country-selection rationale and added an explanatory footnote in the Introduction section.

Please see the footnote on Page 2: “When implementing TALIS 2013, participating countries could add an extra option to their country level study. They could invite all mathematics teachers to participate in an optional survey: the Mathematics Teacher Questionnaire labelled as TALIS-PISA Link. The mathematics teachers were expected to be a subsample of teachers who taught the PISA 2012 students in the same schools. Eight countries opted for this additional study: Australia, Finland, Latvia, Mexico, Portugal, Romania, Singapore, and Spain. The current study is a follow-up of other studies in which we already applied PISA_TALIS linkage data. In a previous study, the measurement model for Mexico appeared being different from any other country. In view of consistency, Mexico was therefore not included in the analyses of the present study. Thus, data from seven countries were used in the subsequent analyses.

For the point 2, "The figures 1-7 are very interesting but their construction is not explained sufficiently", many thanks for this advice. The figures visualize the analysis results. But to make it easier for the reader to understand, we enriched the explanation for figures from line 609:“In the path diagrams of Figure 1-7, the observed variables are represented in boxes, and the latent factors are represented in circles. Significant relationships between factors are depicted with single arrow lines, with the head of the arrow pointing towards the variable being influenced by another factor in the current study. The path coefficients specify the relationships among the specified factors. The curved double arrow lines imply two latent factors being correlated. The residual variance or the measurement errors are the values represented next to the factor with single arrow lines.”

Thanks a lot for your suggestions, the details please see the manuscript.

Reviewer 4 Report

The paper presents quite interesting analysis, however presented in a very hard to follow way. The authors present a lot of data without proper explanation of their meaning - what given  range of given attribute actually mean? Should the values be high/low or similar? What that gives us?

Some of the statements are hard to understand. In 3.4 the authors state that "A value of ICC1 exceeding 0.05 indicates that a multilevel analysis approach is essential to model the data". Meaning? What kind of multilevel analysis is needed? How a single level analysis would look like? In how many cases can we even get ICC1 below 0.05? What is the possible range of ICCs? The same goes with ICC2. Why do we need to aggregate data for ICC2 higher than 0.6?

In the discussion the authors state: " The current findings underscore the importance of understanding socioeconomic status for academic performance and highlight the role of two dimensions of mathematics instructional quality matter for school mathematics performance." What two dimensions are the authors writing about? Why two whereas in the previous paragraphs, especially 4.2, the authors state about three dimensions. Once again those three factors are not stated.

In general the paper should be restructure for greater readability. In current form it is very hard to follow.

Figures 1 -7 are unreadable.

There are some strange English constructions or some words are incorrectly used (e.g.  The current findings underscore the importance). In other cases it's hard to follow that the author had in mind (e.g. "The current research reflects strengths and limitations." - of what/whom?)

The bibliography items with links should provide date of access information. Links can expire, especially on governmental pages.

Author Response

Thank you very much for your suggestions. We response the comments point by point as follows.

Point 1: The paper presents quite interesting analysis, however presented in a very hard to follow way. The authors present a lot of data without proper explanation of their meaning - what given range of given attribute actually mean? Should the values be high/low or similar? What that gives us?

Response 1: We understand this comment! We agree that a more structured explanation is needed. We restructured the text and also added actual numbers/data to the text, so that the reader can follow the rationale.

For example, in the evaluation of model fit from line 318 in Page 7, we added that the value within this range means that the model is acceptable. “To evaluate the model fit, the model-fit indices with the following cut-off values have been taken into account to accept the model: comparative fit index (CFI) ≥ 0.95, root mean square error approximation (RMSEA) ≤ 0.05, and root mean square residual (SRMR) ≤ 0.08 [112].”

Another example, in the explanation of cut-off values between three level of measurement invariance from line 337 in Page 7, we added the meaning of these values: “These recommended cut-off values are then used in the following research to investigate the rejection rates for different degrees of invariance within each level and for various levels of invariance.

Point 2: Some of the statements are hard to understand. In 3.4 the authors state that "A value of ICC1 exceeding 0.05 indicates that a multilevel analysis approach is essential to model the data". Meaning? What kind of multilevel analysis is needed? How a single level analysis would look like? In how many cases can we even get ICC1 below 0.05? What is the possible range of ICCs? The same goes with ICC2. Why do we need to aggregate data for ICC2 higher than 0.6?

Response 2: Thank you very much for pointing this out. The text is too cryptic.The range of ICC1 and ICC2 indicates that the multilevel model is acceptable, significant, and meaningful. In the current study, the “multilevel analysis” is at the student level and school level.

When the values of the ICCs reach to the defined range, it means in our study that socioeconomic status and mathematics achievement can be applied at the school level. We reconstructed this part of the text and elaborated on the meaning of ICCs values. Also, the specific explanation of ICCs of the data for the current study was given. 

Please see Page 8, from line 360: “Two ICC measures can be distinguished. ICC1 represents the proportion of variance of the outcome variable explained by individuals belonging to different groups [118, 119]. In the current analysis, ICC1 captures the variation in SES and mathematics achievement scores can be due to the fact that students belong to different schools. The higher ICC1 is the greater the between-school differences in their students’ SES and Mathematics achievement. ICC,1 in many instances, is used as a measure of school segregation. The ICC2, on the other hand, measures the reliability of the aggregated level variable mean by the proportion observed total variance in group mean scores occurring at the aggregated level [119, 120]. The common guidelines for an acceptable ICC level for multilevel modelling are as the follows. A value of ICC1 exceeding 0.05 indicates that a multilevel analysis is essential and meaningful to adjust for hierarchical data structure. A value of ICC2 value larger than 0.60 implies the reliable aggregation of the within-group data on the group level [118, 119, 121, 122]. The magnitudes of ICC1 and ICC2 enable researchers to assess how clustering of individuals in higher level units affects the observed variation of individual and aggregated scores. Take the ICCs for mathematics performance from Australia as an example. The ICC1 value of 0.31 indicates that 31% of the variance in students’ mathematics achievement scores is due to systematics between-school differences. In contrast, a value of ICCs is 0.90, representing that 90% of the observed total variance in the aggregated school mean mathematics does occur at the school level.”

Point 3: In the discussion the authors state: " The current findings underscore the importance of understanding socioeconomic status for academic performance and highlight the role of two dimensions of mathematics instructional quality matter for school mathematics performance." What two dimensions are the authors writing about? Why two whereas in the previous paragraphs, especially 4.2, the authors state about three dimensions. Once again those three factors are not stated. In general the paper should be restructure for greater readability. In current form it is very hard to follow.

Response 3: Thank you for this comment. Indeed, rereading the current version of the article shows text parts that are very dense and cryptic. We added more information to a restructured version of the text.

We added the specific dimensions – disciplinary climate and supportive climate – in brackets to enrich this sentence. Why these two? Because only these two dimensions of school mathematics instructional quality are significantly related to school mathematics performance.  The dimension of cognitive activation was not significantly related to school performance.

See line 1120-1123: “The current findings underline the importance of understanding socioeconomic status for academic performance and highlight the role of two dimensions of mathematics instructional quality (i.e., disciplinary climate and supportive climate) matter for school mathematics performance.”

The three-factor model has again been stated explicitly at the start of the results section in 4.1; see lines 386-389: “A CFA model of instructional quality was estimated for each country. For five countries (Finland, Singapore, Portugal, Romania, and Spain), the established three factors of schools’ mathematics instructional quality can be identified: disciplinary climate (DC), supportive climate (SC) and cognitive activation (CA).”

And we repeated at several places the specific names of the three factors in the text; see e.g., lines 397-399: “The three-factor model structure (i.e., disciplinary climate, supportive climate, and cognitive activation) did not fit data from Australia and Latvia, where the SC factor could not be identified, …”

The three factors have also been added to the text in the lines 629-632: “Four countries where a three-factor model structure (i.e., disciplinary climate, supportive climate, and cognitive activation) for instructional quality in school mathematics could be confirmed (i.e., …”

Point 4: Figures 1 -7 are unreadable.

Response 4: We fully understand the concerns about the quality of the figures. We checked their quality and in view of a better understanding, extra notes have been added to each figure to help the reader in their interpretation.

Point 5: There are some strange English constructions, or some words are incorrectly used (e.g.  The current findings underscore the importance). In other cases it’s hard to follow that the author had in mind (e.g. "The current research reflects strengths and limitations." - of what/whom?) 

Response 5: Thank you very much for these suggestions. We checked spelling and grammar again and consulted a native EN language speaker.

For instance, “underscore” has been replaced and the sentence now reads as follows (lines 1120-1123): “The current findings underline the importance of understanding socioeconomic status for academic performance and highlight the role of two dimensions of mathematics instructional quality …”

Point 6: The bibliography items with links should provide date of access information. Links can expire, especially on governmental pages.

Response 6: Thank you for your recommendation! The references have been enriched with the requested information.

Please see the details in the new version of manuscript.

Round 2

Reviewer 4 Report

Thank you for your corrections, in my opinion the paper is now much more readable and coherent.IN such state it can be accepted.

I must admit that the number of bibliography items is very high.